# 'Love with Less Salt': evaluation of a sodium reduction mass media campaign in China

Ashish K Gupta  ,[1] Tom E Carroll,[2] Yu Chen,[3] Wenzhuo Liang,[3] Laura K Cobb,[4] Yichao Wang,[5] Juan Zhang,[6] Yeji Chen,[7] Xiaolei Guo,[8] Sandra Mullin,[9] Nandita Murukutla[9]

¹Vital Strategies, New Delhi, India
²Vital Strategies, Sydney, New South Wales, Australia
³Vital Strategies, Beijing, China
⁴Resolve to Save Lives, An Initiative of Vital Strategies, New York, New York, USA
⁵Beijing Haisi Aide Technology Co Ltd, Beijing, China
⁶Chinese Academy of Medical Sciences & Peking Union Medical College, Beijing, China
⁷Anhui Provincial Center for Disease Control and Prevention, Hefei, Anhui, China
⁸Shandong Center for Disease Control and Prevention, Jinan, China
⁹Vital Strategies, New York, New York, USA

**Correspondence to**
Ashish K Gupta;
agupta@vitalstrategies.org

## ABSTRACT

**Objective** This study examines the impact of a salt reduction campaign on knowledge, attitudes, intentions, behaviours and barriers to behaviour change relating to salt consumption in two provinces of China.

**Methods** In 2019, the 'Love with Less Salt' campaign ran on China Central Television and on local television channels in Shandong and Anhui provinces. Data for this study come from two representative household surveys conducted among a sample of adults aged 25–65 years in Shandong and Anhui provinces: precampaign (n=2000) and postcampaign (n=2015). Logistic regression was performed to estimate the effects of the campaign on knowledge, attitudes, intentions, behaviours and barriers to behaviour change.

**Results** Overall, 13% of postcampaign respondents recalled seeing the campaign, and reactions towards the campaign were positive. Postcampaign respondents were more likely to plan to reduce their purchase of foods high in salt than precampaign respondents (OR=1.45, p=<0.05). Campaign-aware respondents were significantly more likely than campaign-unaware respondents to report higher levels of knowledge, attitudes and behaviours regarding salt reduction.

**Conclusions** Findings reveal that salt reduction mass media campaigns can be an effective public health tool to support efforts to reduce salt consumption in China. Continued and sustained mass media investments are likely to be effective in addressing high salt consumption nationwide.

## INTRODUCTION

High sodium intake is a risk factor for hypertension, cardiovascular disease and cerebrovascular disease and morbidity and mortality associated with these conditions.[1–3] The WHO recommends reducing sodium intake to less than 2 g per day (equivalent to 5 g per day of salt). Globally, many adults exceed recommended guidelines, resulting in nearly 1.65 million deaths from cardiovascular disease annually.[4 5] Evidence shows lowering sodium intake to recommended levels can substantially reduce associated risks.[6]

### STRENGTHS AND LIMITATIONS OF THIS STUDY

⇒ For this study, a repeated 'cross-sectional' survey design was used rather than a cohort design, which risks sensitisation bias for a mass media campaign evaluation.
⇒ Multistage probability sampling procedures and data weighting were applied to have robust samples representing the population.
⇒ Frequency of exposure to the campaign was not measured, which can inform the minimum level of exposure required for impact on behaviour change.
⇒ Assessment of campaign impact relied on self-reported measures related to sodium consumption rather than physical measures of actual sodium intake.

Salt is the main source of sodium in diets.[7] In China, average salt intake is among the highest in the world; at 10.5 g per day, it is more than double the amount recommended by the WHO.[8] In 2010, an estimated 441 200 cardiovascular deaths in China—12.8% of the total—were due to high sodium intake, of which 34 500 deaths occurred in Shandong province alone.[9] Home cooking in China contributes to more than three-quarters of sodium consumption (76%),[10] since food preparation often involves adding salt and salty seasonings or condiments.[11 12] Women are the primary preparers of homecooked meals in China and are often the household's nutritional gatekeepers, while men also exert an influence over food purchase and preparation.[13–15] Due to factors such as differences in eating patterns, men consume more salt than women and experience linked diseases at higher rates.[16] Understanding and addressing these gender norms and patterns as they relate to salt intake is important for salt reduction efforts.

Literature on knowledge, attitudes and behaviours regarding sodium intake in China suggests that although people are aware of

the links between high sodium and hypertension, people often associate the use of salt with better tasting food and believe it makes them physically stronger.[12 15] In some cases, this dietary choice also stems from preference for regional cuisine, which in some regions may contain more salt than in others. Shandong's cuisine, for example, is famous for containing more salt and soy sauce than other regional cuisine.[12]

Mass media campaigns have proven to be an effective public health tool for promoting behaviour change and addressing social norms.[17–19] For example, campaigns addressing tobacco use that graphically depict tobacco-related health harms in a personally relevant and engaging way have been found to effectively prompt quitting behaviour to avoid these harms and influence social norms to support changes in smoking behaviour.[20] Undertaking formative research to pretest communication concepts with target audiences has been a key feature of effective tobacco control campaigns. While evidence reflecting the effectiveness of mass media campaigns aimed at reducing salt consumption is limited,[19] it is reasonable to conclude that they have the potential to influence consumption behaviours when strategies are developed and tested through research and implemented within a comprehensive package of sodium reduction interventions.[19 21 22]

Effective mass media campaigns convey both essential health information and persuasive, emotionally appealing stories and images.[23 24] Research on sodium intake in China suggests that, to be effective, mass media campaigns need to show people how they consume excessive sodium (eg, salt added during home cooking) and how their association of salt with better taste and physical strength can perpetuate unhealthy eating habits.[15]

Based on this evidence, in 2019, the Centers for Disease Control and Prevention of both Shandong and Anhui provinces, with technical assistance from Vital Strategies, a global public health organisation, and the Resolve to Save Lives initiative, which supports governments in both provinces in sodium reduction efforts, developed and implemented an evidence-based mass media campaign targeting household salt reduction. This was a first-of-its-kind research-based mass media campaign to target salt consumption in China and complemented the 2019 Healthy China 2030 Action Plan, which recommends the 5 g of salt per person per day limit. The objective of this study was to determine whether the campaign led to changes in knowledge, attitudes, intentions and behaviours around salt use in Shandong and Anhui provinces by conducting precampaign and postcampaign cross-sectional surveys.

## METHODS
### A mass media campaign in Shandong and Anhui provinces
#### Campaign development
Based on advice from provincial programme team members, the primary target audience designated for the campaign were women who were primary food preparers for their household as they exerted critical influence on sodium consumption and family health. From August to September 2018, 13 focus groups adopting a methodology of quantitative ratings and group discussions were undertaken with primary food preparers (nine groups of women 25–65 years) and with the secondary audience of other influencers of household cooking (four groups of men 25–55 years) in each province. Participants were recruited according to age, gender and geographical specifications in line with the campaign target audience by an independent market research agency, which conducted the focus groups. The objective of this formative research was to test the relative effectiveness of different messages and message styles to motivate primary food preparers to reduce salt in their cooking and to encourage influencers to support this change. These communication concepts were presented in animatic videos and included the depiction of a happy family providing positive encouragement for cooking with less salt, demonstration of how to replace salt in cooking but still maintain taste, presentation of lower salt cooking by a celebrity chef and a testimonial story of a man who experienced hypertension and stroke due to excessive salt consumption.

The research findings indicated that while it was relatively easy to communicate information about limiting salt consumption, engaging and motivating people to recognise the serious consequences of excessive salt consumption and the importance of changing their behaviour in relation to salt use in cooking required communication at both an emotional and informational levels. The findings showed that the testimonial concept was most effective in generating this engagement and motivation but lacked sufficiently clear direction about how to avoid harms from excessive salt consumption. Informed by these findings, the campaign strategy and production of the campaign public service announcement (PSA) for television broadcast focused on a personal narrative of harm from excessive salt consumption, complemented by expert advice about reducing salt intake presented by a medical doctor. The 'Love with Less Salt' campaign's 30 s PSA featured the story of a man who developed hypertension and suffered a stroke due to his ongoing consumption of salty food, despite his wife's warnings. The formative research demonstrated that an important feature of the script's impact for both female and male participants was the husband's introspection and regret about the ongoing burden that he was now imposing on his family, and this was carefully embedded in the PSA. In the final script of the PSA, the doctor identifies excessive salt intake as the main cause of hypertension, which increases the risk of stroke and heart disease, and urges limiting salt consumption to no more than 5 g/day, stating: 'For the health of your family, use less salt when you cook'.

## Campaign dissemination

The 'Love with Less Salt' PSA was broadcast on national, provincial and local television stations through a paid media plan. The PSA aired September–October 2019 in Shandong and Anhui provinces, with a combined population exceeding 160 million people, supporting broader campaign efforts in the two target provinces, including billboards, posters and community education activities undertaken by the provincial Centers for Disease Control and Prevention. China Central Television also broadcast the PSA across the country throughout September 2019.

## Evaluation study design

To evaluate campaign effectiveness, precampaign and postcampaign cross-sectional representative household surveys were undertaken. Surveys were conducted in-person by an independent research agency hired by vital strategies. The precampaign survey was conducted, leading up to the campaign in July 2019; the postcampaign survey was conducted following the campaign from November 2019 to January 2020.

## Sampling method and sample size

A multistage probability sampling procedure was used with stage-specific sampling frames to select participants for the surveys in Anhui and Shandong provinces. In each province, cities were selected to ensure representation across different city tiers; the selection of districts and subdistricts corresponded accordingly. In the case of rural areas, towns were selected. Finally, households and respondents within households were selected. A computer-aided personal interviewing random selection programme was used to select one respondent from among the multiple eligible respondents in a household. Reflecting the campaign target audience, respondents' eligibility criteria included being adults aged 25–65 years who influence food preparation decisions in the home. Quotas were set for 75% women who were primary food preparers in the home and 25% men who influence food preparation, including salt use by the primary food preparer.

A target sample of 1000 people per province was designed at a 95% level of confidence, 5.5% margin of error and design effect of 1.2. Accordingly, a total sample of 2000 and 2015 eligible adults was included in precampaign and postcampaign surveys, respectively.

## Survey questionnaire and measures

A pretested questionnaire was administered in-person in Chinese using a computer-aided personal interviewing device to avoid any selection biases. Questions in the survey were ordered to minimise order effects and related biases. The duration of the interviews was approximately 30 min. The key measures that were used in the survey are described below.

*Sociodemographic information*: asking age, gender, income, occupation, education, marital status, media consumption habits, location and whether the respondent was a parent of children under 16 (table 1).

*Campaign awareness*: showing respondents still images depicting key scenes from the television PSA and asking whether the respondent had seen the PSA before. Those who recalled seeing the campaign PSA were categorised as 'campaign aware'. All other respondents were categorised as 'campaign unaware'.

*Reactions to the campaign*: using a 5-point Likert scale to measure levels of agreement among campaign-aware respondents on a battery of 16 statements about responses to the PSA, for example, the ad 'taught me something new', 'made me stop and think' and 'motivated me to discuss the ad with others' (see table 2 for the complete list of statements).

*Knowledge* about salt consumption: asking a series of 'yes' or 'no' questions that probed respondents' knowledge about the recommended daily salt intake and understanding of related health risks.

*Attitudes* towards salt consumption: measuring respondents' agreement (on five-point Likert scales) with a series of statements, such as: 'It is very important to reduce salt in cooking' or 'making efforts to use salt alternatives is worthwhile given the health benefits' (see table 3 for the complete list of statements).

*Behaviour* related to salt consumption: using a series of 'yes' or 'no' questions related to cooking, eating and recognising 'too much' salt.

*Behavioural intentions* related to salt consumption: using a series of 'yes' or 'no' questions and Likert-scale responses such as intention to reduce purchase of foods very high in salt.

*Barriers* to salt reduction: asking respondents to indicate agreement (using a 5-point Likert scale) to a list of potential barriers for reducing salt such as being more concerned about other ingredients (see table 4 for the complete list of statements).

## Data analysis

Data were analysed using SPSS V.25. The proportion of respondents indicating 'agree' and 'strongly agree' on the 5-point Likert scale were aggregated for analysis and reporting. Comparisons between proportions in the precampaign and postcampaign surveys and between respondents who were aware and unaware of the campaign were conducted using $\chi^2$ tests. Logistic regression was performed to examine the association between respondents' awareness of the campaign and their knowledge, attitudes, behavioural intentions and behaviours regarding salt reduction as well as barriers to changing behaviour. Covariates included gender, age, primary caregiver status, marital status, education, occupation, income and media consumption habits (frequency of watching television and frequency of internet use).

## Patient and public involvement

Patients or the public were not involved in the design, conduct, reporting or dissemination plans of our research.

**Table 1** Demographic characteristics of respondents to the precampaign and postcampaign surveys (%)

|  | Precampaign | Postcampaign |  |  |
|---|---|---|---|---|
|  | Total | Total | Unaware | Aware |
|  | (n=2000) | (n=2015) | (n=1744) | (n=271) |
| Location |  |  |  |  |
| Tier 2 city | 40 | 40 | 43a | 23b |
| Tier 3 city | 30 | 30 | 27a | 48b |
| Tier 4 city | 30 | 30 | 30 | 29 |
| Women (primary food preparers) | 75 | 75 | 75 | 74 |
| Men (influencers) | 25 | 25 | 25 | 26 |
| Age in completed years (mean) | 41.6a | 48.6b | 48.7 | 48.3 |
| Education |  |  |  |  |
| Middle school and below | 42a | 55b | 57a | 43b |
| High school | 23a | 19b | 31a | 39b |
| College and higher | 36a | 26b | 12a | 18b |
| Occupation |  |  |  |  |
| Employed | 58a | 52b | 52 | 56 |
| Homemaker | 24a | 20b | 20 | 19 |
| Retired | 10a | 20b | 21 | 16 |
| Unemployed | 7a | 7b | 7 | 9 |
| Student | 0.5a | 1.1b | 1 | 1 |
| Household income (annual) |  |  |  |  |
| Up to 30K RMB | 27a | 23b | 23 | 25 |
| 31K–70K RMB | 28a | 34b | 35a | 27b |
| 71K–110K RMB | 22 | 21 | 21a | 26b |
| Above 110K RMB | 13 | 12 | 12 | 16 |
| Parent of children below 16 years | 45a | 30b | 28a | 40b |
| Married/living together | 90a | 87b | 87a | 91b |
| Watch television every day | 48a | 43b | 42a | 51b |
| Use internet many times a day | 50a | 44b | 44 | 44 |

Values in the same row with different subscripts 'a' and 'b' are significantly different at p<0.05 in the two-sided test of equality for column proportions.

## RESULTS

### Demographic characteristics of respondents

Table 1 presents demographic characteristics of the precampaign and postcampaign samples. There were significant differences between precampaign and post-campaign respondents across several measures. Respondents to the postcampaign survey were significantly older (48.6 years old vs 41.6 years old), less educated (26% vs 36% attended college and higher) and more likely to be retired (20% vs 10%) than those to the precampaign survey. Postcampaign respondents were significantly less likely to watch television (43% vs 48%) or use the internet (44% vs 50%) than those in the precampaign survey.

### Campaign awareness and recall

A significantly higher proportion of those who were aware of the campaign had completed high school, college or higher education (57%) than those who were unaware (43%). Those who were aware of the campaign

were also more likely to watch television everyday (51% vs 42%). The top three messages recalled were: 'reducing salt consumption is good for health' (78%); 'excess salt intake leads to hypertension/heart attack/stroke' (53%); 'limit salt intake to 5 g or less per day per person' (32%). The campaign slogan 'too much salt will lead to stroke and even death' was recalled by 53% of campaign-aware respondents.

### Reactions to the campaign

The majority of respondents who recalled the campaign agreed it was believable (93%), relevant (93%), taught them something new (88%) and made them stop and think about excessive salt intake (82%; see table 2). The majority of campaign-aware respondents reported it made them feel concerned about the effects of eating too much salt on their family's health (92%) and on their own health (90%). In addition, 80% agreed that the campaign made them feel concerned about too much salt being

**Table 2** Reactions to the campaign among campaign-aware respondents in post-campaign survey (%)

| Self-reported reactions to PSA | % agree (n=271) |
|---|---|
| Was believable | 93 |
| Was relevant to me | 93 |
| Taught me something new | 88 |
| Made me stop and think | 82 |
| Made me feel uncomfortable | 22 |
| Made me concerned about too much salt being used in cooking in my household | 80 |
| Made me feel concerned about effects of eating too much salt on my health | 90 |
| Made me feel concerned about effects of eating too much salt on my family's health | 92 |
| Made me want to use less salt in my cooking | 95 |
| Made me want to eat less salt | 95 |
| Made me more confident to cook with less salt/ encourage less salt in cooking in my family home | 95 |
| Made me more supportive of government action to reduce salt consumption in my country | 96 |
| Is useful for public education | 97 |
| **Intended interpersonal communication** | |
| Motivated me to discuss the ad with others | 73 |
| I would like others to see this ad | 97 |
| I would like my family members to see this ad | 96 |

PSA, public service announcement.

used in household cooking. Almost all campaign-aware respondents (95%) reported it made them more confident to cook with less salt and encourage less salt in home cooking, and to want to use less salt in cooking and eat less salt. Similarly, almost all campaign-aware respondents agreed it made them more supportive of government action to reduce salt consumption nationwide (96%) and that it is useful for public education (97%). Furthermore, almost three-quarters of campaign-aware respondents agreed that they would discuss the campaign with others (73%).

## Knowledge and attitudes about salt intake
### Changes between the precampaign to the postcampaign surveys
There were significant differences in key measures of knowledge and attitudes regarding salt intake between the precampaign and postcampaign surveys (see table 3). A higher proportion of postcampaign respondents agreed the 'recommended daily salt intake should be less than 5 g per day' (40%) than precampaign respondents (34%). Nearly 9 in 10 (89%) postcampaign respondents 'worried that too much salt consumption in their family can have very serious consequences', compared with 85% of precampaign respondents. The proportion of respondents considering the health risks of too much salt was significantly higher in the postcampaign survey compared with the precampaign survey (54% vs

45%). Similarly, a greater proportion of postcampaign respondents (43%) considered how to reduce family salt consumption, compared with precampaign respondents (38%). Certain parameters were lower in the post-campaign survey, such as, knowledge that 'eating high salt/salty seasoning is harmful', and 'sauces like soy, fish sauces, oyster sauces being high in salt' and perception that the 'use of salt alternatives is a safe way of reducing sodium intake'; however, these were significantly higher among campaign-aware respondents than campaign-unaware respondents.

### Impact of campaign awareness within the post-campaign survey
Knowledge of the health consequences of excessive salt intake was significantly higher among those aware of the campaign. As described in table 3, the proportion agreeing that the 'recommended daily salt intake should be less than 5 g per day' was significantly higher among campaign-aware respondents (49%) compared with campaign-unaware respondents (39%). While the perception that 'eating high levels of salt or salty seasoning will be very harmful for health' was significantly higher among campaign-aware respondents, these perceptions were generally lower in the postcampaign survey (38% vs 44%). Campaign-aware respondents were significantly more likely to agree that consuming high levels of salt has negative health consequences, including hypertension (93% vs 85%), heart attack (69% vs 60%), stroke (65% vs 58%) and bone health (51% vs 39%). Postcampaign respondents who recalled the campaign were significantly more likely to believe that 'it is very important to reduce salt in cooking' compared with those who did not recall the campaign (72% vs 53%), and that 'use of salt alternatives is a safe way of reducing sodium intake' (30% vs 25%). Similarly, among postcampaign respondents, the belief that using salt alternatives has positive health benefits was higher among those who recalled the campaign (84% vs 77%).

## Intentions, behaviours and barriers related to reducing salt consumption
### Changes between the precampaign and the postcampaign surveys
Table 4 presents the findings on intentions and behaviours related to reducing salt use as well as barriers to doing so. Immediate intentions to reduce high-salt food purchases were higher among postcampaign respondents than precampaign respondents (86% vs 80%). However, the level of confidence in one's ability 'to control salt consumption while maintaining the appealing taste of food' was lower among postcampaign respondents (80% vs 85%). While the proportion that reported 'measuring the amount of salt added while cooking' was significantly higher in the postcampaign survey than in the precampaign survey (50% vs 46%), the proportion that reported 'adding less salty processed foods now than 3 months ago' was lower in the postcampaign survey (16% vs 26%).

The most cited barrier to salt reduction—a greater concern for the negative health consequences of

**Table 3** Knowledge and attitudes about salt intake, precampaign and postcampaign (%)

| | Precampaign | Postcampaign | Postcampaign | Postcampaign | |
| | Total | Total | Unaware* | Aware* | Adjusted OR† |
|---|---|---|---|---|---|
| | (n=2000) | (n=2015) | (n=1744) | (n=271) | |
| **Knowledge** | | | | | |
| Recommended daily salt intake should be<5 grams/day | 34.0a | 40.1b | 38.8a | 48.8b | 1.58** |
| Eating high salt/salty seasoning will be very harmful | 43.6a | 38.3b | 36.2a | 51.7b | 1.69* |
| Consuming high levels of salt lead to: (% yes) | | | | | |
| Hypertension | 87.6 | 85.7 | 84.6a | 92.5b | 1.87* |
| Heart attack | 59.6 | 60.8 | 59.5a | 69.3b | 1.51* |
| Stroke | 59.9 | 58.9 | 57.9a | 65.2b | 1.32 |
| Harmful for bones | 45.5a | 40.4b | 38.7a | 50.6b | 1.64* |
| Knowledge about foods high in salt: (% definitely) | | | | | |
| Seasoning like stock cubes and powders | 18.7a | 22.8b | 21.3a | 32.1b | 1.72* |
| Sauces like soy, fish sauce, oyster sauce, etc. | 62.2a | 58.0b | 55.7a | 72.3b | 2.01* |
| **Attitudes** | | | | | |
| About salt reduction: | | | | | |
| It is very important to reduce salt in cooking (% yes) | 64.1a | 55.6b | 53.2a | 71.6b | 1.96* |
| Use of salt alternatives is a safe way of reducing sodium intake (% definitely) | 29.2a | 25.6b | 24.9a | 30.3b | 1.43** |
| (% Strongly agree and somewhat agree) | | | | | |
| Making efforts to use salt alternatives is worthwhile for the health benefits | 76.5 | 77.9 | 76.9a | 83.8b | 1.43** |
| I am worried that too much salt consumption in my family can have very serious consequences | 84.8a | 89.3b | 88.7a | 93.4b | 1.32 |
| It's very difficult to control salt consumed in a day | 61.3 | 61.8 | 63.4a | 51.7b | 0.67* |
| I am worried reducing salt in cooking may have negative effects on family's health | 30.7a | 34.0b | 33.8 | 35.8 | 1.11 |
| Reducing salt in cooking is not a priority for me | 35.8a | 49.5b | 49.8 | 48 | 0.98 |
| Reducing salt in cooking is something I support | 93.3a | 95.5b | 95.2 | 97.4 | 1.81 |
| Reducing salt consumption in my family is a positive way to maintain good health | 94.0a | 96.0b | 95.8 | 97.4 | 1.62 |
| Considered following in last 3 months: (% always/often) | (n=2000) | (n=2015) | (n=1633) | (n=257) | |
| Amount of salt/salty sauces added while cooking | 12.6a | 22.4b | 21.4a | 28.8b | 1.36** |
| Amount of salt added to food before eating | 7.9a | 15.9b | 14.4a | 25.8b | 2.08* |
| Health harms from consuming high salt | 45.2a | 54.0b | 52.0a | 66.8b | 1.82* |
| Ways to reduce salt the family consumes | 37.7a | 43.0b | 41.2a | 54.6b | 1.64* |

Values in same row with different subscript 'a' and 'b' are significantly different at p<0.05 in two-sided test of equality for column proportions.
*Comparisons are currently based on bivariate analysis alone and do not control for potential confounders. Hence, any significant differences between 'aware' and 'unaware' groups must be interpreted with caution. *significance at 0.01,
**significance at 0.05.
†Adjusted for gender, age, primary caregiver, marital status, education, occupation, income, television watching frequency and internet use frequency.

**Table 4** Intentions, behaviour and barriers related to reducing salt consumption, precampaign and postcampaign (%)

| | Pre-campaign | Post-campaign | | | Adjusted OR† |
|---|---|---|---|---|---|
| | Total | Total | Unaware* | Aware* | |
| | (n=2000) | (n=2015) | (n=1744) | (n=271) | |
| **Immediate intentions** (% strongly/somewhat agree) | | | (n=1633) | (n=257) | |
| To reduce purchase of foods very high in salt | 79.6a | 85.8b | 85.1a | 90.0b | 1.45* |
| **Confidence in controlling salt use** (% confident) | | | (n=1633) | (n=257) | |
| Reduce salt in cooking to avoid family's salt intake exceeding the recommended levels | 82.2 | 82.9 | 82.1a | 87.9b | 1.59** |
| Maintain appealing taste for food for my family with reduced salt in cooking | 84.7a | 80.3b | 79.1a | 87.5b | 2.07* |
| **Behaviour related to purchase** (% yes) | | | (n=1633) | (n=257) | |
| Look for salt content on food labels | 23.5 | 24.6 | 22.8a | 36.2b | 1.8* |
| Buy low-salt alternatives | 21.0 | 22.9 | 19.8a | 42.8b | 2.88* |
| Buy low-sodium salt | 33.8 | 35.1 | 32.2a | 53.5b | 2.35* |
| **Behaviour while cooking** (% yes) | | | (n=1633) | (n=257) | |
| Measure amount of salt | 45.7a | 49.9b | 47.8a | 63.4b | 1.84* |
| Use salt reduction measuring spoon | 32.4 | 31.4 | 29.1a | 45.5b | 2.01* |
| Replace salt using onions, garlic, ginger | 17.1a | 23.2b | 22.6 | 27.2 | 1.30 |
| Add less salt than 3 months ago | 20.5 | 19.3 | 18.0a | 27.2b | 1.56* |
| Add less salty processed foods than 3 months ago | 26.3a | 16.1b | 14.3a | 27.2b | 1.94* |
| Add less sauce or seasoning than 3 months ago | 17.7 | 15.8 | 14.1a | 26.5b | 2.01* |
| **Behaviour while eating** (% yes) | | | (n=1633) | (n=257) | |
| Limit consumption of processed foods | 40.1a | 56.1b | 54.9a | 63.8b | 1.44* |
| Avoid eating outside foods | 43.7a | 59.7b | 58.8 | 65.3 | 1.27 |
| Consume less salt overall | 17.5 | 18.9 | 17.5a | 27.7b | 1.65* |
| **Barriers to salt reduction** (% strongly agree/agree) | | | | | |
| I am more concerned about other ingredients | 70.3a | 61.4b | 61.8 | 58.7 | 0.88 |
| Can't tell how much salt is in the foods I like to eat | 58.2a | 65.2b | 67.7a | 49.1b | 0.48* |
| Can't tell how much salt on food label is too much | 41.4 | 43.7 | 45.0a | 35.1b | 0.79 |
| My doctor suggested not to reduce my salt intake due to my certain disease or current treatment | 38.1a | 28.6b | 27.7a | 34.7b | 1.4** |

Values in same row with different subscript 'a' and 'b' are significantly different at p<0.05 in the two-sided test of equality for column proportions.
*Comparisons are currently based on bivariate analysis alone and do not control for potential confounders. Hence, any significant differences between 'aware' and 'unaware' groups must be interpreted with caution.
†Adjusted for gender, age, primary caregiver, marital status, education, occupation, income, television watching frequency and internet use frequency.

ingredients such as fat, carbohydrates, or sugar rather than that of salt—was reported by fewer respondents in the postcampaign survey than in the precampaign survey (61% vs 70%). More than two-fifths of precampaign and postcampaign survey respondents reported not knowing how much salt on a food label is too much (41% vs 44%) and limited low-salt food options while shopping or dining (44% vs 45%). The proportion of respondents who said their doctor advised against reducing salt was lower in the postcampaign survey compared with the precampaign survey (29% vs 38%).

### Impact of campaign awareness within the postcampaign survey

As shown in table 4, intentions to reduce high-salt food purchases were higher among respondents who recalled the campaign compared with those who did not (90% vs 85%). A significantly higher proportion of campaign-aware respondents reported confidence in reducing salt during food preparation to limit family consumption (88% vs 82%) and confidence in preserving taste while reducing salt (86% vs 79%). Considerably more campaign-aware respondents reported that they had decreased their overall salt consumption in the previous 3 months than unaware respondents (28% vs 18%).

Significantly more campaign-aware respondents reported 'measuring the amount of salt added while cooking' (63% vs 48%). Almost two-thirds of campaign-aware respondents reported limiting consumption of processed food and avoiding eating outside foods (64% and 65%) compared with a little more than half of those not aware of the campaign (55% and 59%). A significantly higher proportion of respondents who recalled the campaign reported looking for salt content on food labels (36% vs 23%), buying low-salt alternatives (43% vs 20%) and buying low-sodium salt (54% vs 32%) compared with those not aware of the campaign. More than two-thirds of respondents unaware of the campaign reported they could not tell how much salt is in the foods they like to eat, compared with only one-half of respondents who recalled the campaign (68% vs 49%). Similarly, reporting not knowing how much salt on a food label is too much was significantly lower among respondents who were aware of the campaign compared with those who were unaware (35% vs 45%).

### DISCUSSION

Consumer education and awareness are essential components of sodium reduction strategies around the world.[25] This paper describes the first comprehensive study to evaluate the effect of a mass media campaign on changes in knowledge, attitudes, intentions, behaviours and barriers regarding reducing salt consumption in China, where an excessively salty diet is a major public health challenge and contributor to significant morbidity and mortality. The 'Love with Less Salt' campaign described in the paper was conducted in China's Shandong and Anhui provinces, where salt consumption is particularly high.

The study findings demonstrated that the 'Love with Less Salt' campaign performed as intended. Though overall awareness of the campaign may appear relatively modest (13% overall; Shandong 16%, Anhui 11%), the campaign reached approximately 12.6 million people in Shandong and Anhui provinces, based on campaign recall findings. The observed lower levels of watching television and internet use among postcampaign respondents may have resulted in lower recall findings. Among those who recalled the campaign, it was very well received; respondents found it believable, relevant and that it made them stop and think about excessive salt use. Almost all campaign-aware respondents (95%) reported it made them more confident to cook with less salt and encourage less salt in home cooking. Approximately three-quarters of respondents reported that seeing the campaign motivated them to discuss it with others, potentially expanding the campaign's reach and influence.[18] Despite modest recall of the campaign, significant improvements across several indicators of change (knowledge and attitudes) were observed between the precampaign and postcampaign surveys. The higher levels of knowledge of the recommended daily salt intake, the increased recognition that too much salt consumption can have serious consequences on one's family and the heightened awareness that reducing salt consumption is a positive way to maintain good health were consistent with a positive campaign impact, as were the observed improvements in intentions, behaviours and barriers regarding lower salt consumption.

While the changes between the precampaign and postcampaign surveys may have been influenced to some degree by concurrent activities beyond the campaign, the comparison of campaign-aware and campaign-unaware respondents in the postcampaign survey suggests an independent impact of the campaign. After controlling for potential confounders, the data showed that those who were aware of the campaign were significantly more likely than those who were unaware to demonstrate higher levels of knowledge, more positive attitudes and improved intentions and behaviours related to reducing salt consumption. In line with specific campaign objectives, campaign-aware respondents were significantly more likely than unaware respondents to understand the harms of excess salt intake, to recognise the importance of reducing salt in cooking and dining and to understand effective approaches for doing so, such as limiting purchases of food high in salt, looking for salt content on food labels and buying low-salt alternatives and low-sodium options. Campaign-aware respondents had significantly higher levels of confidence in controlling salt intake in cooking and were more likely to report changing their purchasing, cooking and eating behaviours compared with those who were unaware. The campaign's high effectiveness ratings and its positive impact on sodium reduction knowledge and intentions among campaign-aware respondents validate the rigorous formative research process undertaken to guide the campaign communication strategy and PSA production.

Finally, this study identified some key barriers to behaviour change in reducing salt intake. In both the precampaign and postcampaign surveys, a majority of respondents reported that they were more concerned about other ingredients in foods, such as sugar and fat, than they were about salt. However, this reported barrier was significantly higher among precampaign respondents, which suggests that the campaign was effective in raising concerns about salt. Future sodium reduction campaigns should continue to highlight the harms of high salt consumption in comparison with other food ingredients that affect health.

In both surveys, a large majority of respondents reported they could not tell how much salt is in the foods they like to eat. Just as a mass media campaign like 'Love with Less Salt' can educate and motivate behaviour change, there is a clear role within a comprehensive sodium reduction strategy for food policy measures to facilitate that behaviour change, including enacting food policies that ensure labelling, can be easily understood by consumers and products available for purchase are lower in salt.

This study builds on literature showing the effectiveness of mass media campaigns in addressing behavioural risk factors, including high salt intake, that can lead to disease and premature death.[18] It shows that mass media campaigns can play a crucial role in improving awareness and changing knowledge, attitudes, preferences and behaviours regarding nutrition and diet.[18 26–29] The study replicates findings from other countries by demonstrating that the salt consumption campaign performed as intended[30] and contributes to emerging literature in China about reducing salt consumption[17] and designing future such campaigns.

### Study limitations

While the repeated cross-sectional survey design of the evaluation precluded that the assessment of individual-level change resulting from the intervention and can represent a study limitation, adopting a cohort design for a campaign evaluation risk sensitisation bias. Respondents for the two surveys were selected randomly to represent the overall target population of each province; however, significant demographic differences were evident between the two samples and needed to be controlled for in data analysis. There are also limitations to the quasiexperimental study design which was necessitated since the campaign exposure was across whole provincial populations and no control group comparison was possible.

The study did not measure frequency of exposure to the campaign. Estimating the relationship between frequency of exposure to the campaign and behaviour change can inform the minimum level of exposure required for impact, which can be useful for developing cost-effective campaigns, especially in low-income countries with limited resources. It is also possible that respondents who were already interested in reducing salt consumption may have been more likely to recall the campaign.

This assessment of campaign impact relied on self-reported measures related to sodium consumption rather than physical measures of actual sodium intake. Some studies have found that despite seeing changes in knowledge, attitudes and practices, no changes in actual sodium intake were seen.[31] Physical measures like urine levels can be used to assess longer term change in the population's sodium consumption in line with China's sodium reduction goals.

### CONCLUSION

In China, a typical high-salt diet is a major public health challenge that must be addressed with comprehensive, evidence-based interventions and policies. Findings from this study suggest that a mass media salt reduction campaign can be an effective public health tool to encourage reduced salt consumption in China. The improved levels of salt reduction knowledge, attitudes, intentions and behaviours and reduced barriers to limiting salt consumption observed among those who recalled the campaign reflect the campaign effectiveness.

While the findings from this campaign evaluation are very encouraging, sustained mass media campaigns will be required to consolidate these gains across Chinese households. Applying best practice principles of conducting thorough research to develop effective campaign communication strategies and materials, careful targeting of mass media broadcasts, and adequate funding to reach wide audiences will be essential to maximise the impact of such campaigns.

**Acknowledgements** The authors gratefully acknowledge the Shandong and Anhui Centres for Disease Control and Prevention; the research teams from the Chinese Academy of Medical Sciences and Peking Union Medical College who undertook fieldwork and preliminary data analysis for the surveys; Jun Cai, MD and Chief of Hypertension Centre, Fu Wai Hospital, who kindly provided a shooting venue for the PSA and portrayed a doctor in it. The authors also thank colleagues at Vital Strategies for their reviews and edits to this paper, including Julia Berenson, PhD; Genine Babakian, Karen Schmidt, MPH and Hana Raskin, M.S.

**Contributors** AKG: Study and questionnaire design; oversight of evaluation field activities; design and direction of data analysis; data interpretation; literature review; writing of this article, responsible for the overall content as the guarantor. TEC: Inputs to questionnaire design; oversight of the field activities; help in writing and analysis; data interpretation; technical assistance for the campaign; literature review; contribution to questionnaire development and writing; coordination with stakeholders. YC, WL: Review and overall local guidance of this evaluation. LKC: input into questionnaire, overall input into media campaign strategy including implementation and evaluation. YW: Survey coordination and conduct of the analysis. JZ: Supervision of conduct of survey and analysis. YC, XG: Strategic planning and implementation of mass media and implications of evaluation findings. SM: Review and overall guidance of this evaluation. NM: Conception, study design and direction of data analysis; data interpretation; supervision in writing this article. All authors contributed to manuscript revision and read and approved the submitted version.

**Funding** This analysis was conducted on behalf of Resolve to Save Lives. Resolve to Save Lives is funded by grants from Bloomberg Philanthropies (50 732 (DAF) and 50 731.01 (BFF); the Bill and Melinda Gates Foundation/Gates Philanthropy Partners/Chan Zuckerberg Initiative (OPP1175906). The authors have not received any additional remuneration to write this article.

**Competing interests** None declared.

**Patient and public involvement** Patients and/or the public were not involved in the design, or conduct, or reporting, or dissemination plans of this research.

**Patient consent for publication** Not applicable.

**Ethics approval** The study was evaluated and approved under applicable U.S. regulations related to human subjects research by the Vital Strategies' human protections administrator (approval number 2019/15). The proposed evaluation is exempt HSR within the meaning of the Common Rule because it is: (2) Research that only includes interactions involving educational tests (cognitive, diagnostic, aptitude, achievement), survey procedures, interview procedures, or observation of public behaviour (including visual or auditory recording) and any disclosure of the human subjects' responses outside the research would not reasonably place the subjects at risk of criminal or civil liability or be damaging to the subjects' financial standing, employability, educational advancement or reputation exempted this study Participants gave informed consent to participate in the study before taking part.

**Provenance and peer review** Not commissioned; externally peer reviewed.

**Data availability statement** Data are available upon reasonable request. The data can be made available to other researchers following publication and after the researchers sign a contract with Vital Strategies about data use.

**ORCID iD**
Ashish K Gupta http://orcid.org/0000-0003-1735-3323

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
