## [Reviewer comments · BMJ Open]

ARTICLE DETAILS

TITLE (PROVISIONAL)	"Love with Less Salt": Evaluation of a Sodium Reduction Mass Media Campaign in China
AUTHORS	Gupta, Ashish; Carroll, Tom; Chen, Yu; Liang, Wenzhuo; Cobb, Laura; Wang, Yichao; Zhang, Juan; Chen, Yeji; Guo, Xiaolei; Mullin, Sandra; Murukutla, Nandita

VERSION 1 – REVIEW

REVIEWER	LI, Wei Tianjin Centers for Disease Control and Prevention, Institute of Non-Communicable Diseases Control and Prevention
REVIEW RETURNED	24-Nov-2021

GENERAL COMMENTS	The work presented for review raises a very important topic-reducing salt intake. It is important in the world, especially in China, which was the highest salt intake country in the world. High salt intake can lead to high morbidity and mortality of NCDs, such as hypertension, kidney diseases, ischemic heart disease and etc.. Media investments should be one of the essential way in reducing salt intake Campaign. This manuscript examined the impact of a salt reduction campaign on knowledge, attitudes and behaviors in relation to salt consumption in two provinces of China. It provides guidance for the full implementation of such media campaigns. It is a very meaningful research. I enjoyed reading your study I appreciate the importance of this study. I wrote my concerns to the editor and I am sure s/he will communicate with you. Minor comments: The comparisons between proportions which were conducted using chi-square tests, whether chi square values can be listed in the table. In the "Data analysis" section, why the variate of Location was not be included in the covariates? Can the frequency and the ways (TV, billboards, posters and community education activities.....) to get the Campaign information be evaluated in the study? Page 9 Line 23, "using salt alternatives is a safe way to reduce sodium intake" is a little different with the description in the Table 3. There are some similar expressions. Please try to keep them consistent for the convenience of readers.
--

REVIEWER	Robles, Brenda Division of Chronic Disease and Injury Prevention, Los Angeles County Department of Public Health
REVIEW RETURNED	26-Dec-2021

GENERAL COMMENTS

General comments: The present study sought to examine the impact of a salt reduction sodium campaign on the knowledge, attitudes, and behaviors of residents of two provinces in China. This study is novel in that few manuscripts have focused on sodium reduction, especially on the impact of health marketing campaigns focused on sodium reduction. However, the manuscript can be significantly improved as follows:

Abstract

Objective- is the campaign about salt or sodium reduction? Throughout the main text, “salt” and “sodium” were used interchangeably. Also, is this study truly just seeking to examine the impact of the campaign when the majority of the sample was not exposed to it? Would reconsider the study aims to more accurately capture what is presented in the tables.
Results- do the authors mention purchase of “foods” and not “goods”? It would be better to report CIs instead of p-value.
Conclusion- What do the authors mean by encouraging “positive change”? This is a bit vague, would consider eliminating or using more specific language.

Article Summary

The first bullet is confusing, not sure what the authors mean

Introduction

Found that the Introduction, as currently framed, was a bit disjointed from the rest of the text (i.e., Methods, Results, Discussion). It does not fully set the stage for the study purpose (which is currently unclear). For example, in the Methods section, the authors appear to mention that the campaign was tailored to women because they are the nutritional gatekeepers of the household. If this is the case, the authors should mention this early on in the Introduction and cite pertinent literature to support that women in China are nutritional gatekeepers of the household. Also, there should be some mention as to why the two provinces were assessed; is it because the authors suspect there may be regional differences in sodium knowledge, attitudes, and behaviors? The Introduction could also be improved in terms of grammar and syntax. Moreover, sodium and salt are used interchangeably (and throughout the text). It might help to provide a definition in the Introduction distinguishing between the two terms. Additional suggestions are the following:
-Line 23: What do authors mean by “physical strength”?
-Line 25: The authors should consider providing some examples of “traditional cuisine” dietary preferences in China. Are these “traditional cuisines” high in sodium/salt? If so, would make that more explicit.
-Lines 26-27: Would split up this sentence. Something along the lines of, “There is evidence that mass media campaigns can be an effective public health tool for changing social norms and promoting behavior change. For example, studies focused on tobacco cessation have found [XYZ]”
-Line 49: Would clarify what type of organization is “Vital Strategies.” Is it a health marketing firm? Similarly, what kind of initiative is “Resolve to Save Lives”?
-Lines 47-56: This last paragraph can be significantly improved for clarity. While the authors describe the purpose of the campaign, the study purpose/main research question (and corresponding study aims) are unclear. The authors should also briefly describe the

	research methodology (e.g., cross-sectional, pre-post study carried out in two regions of China?) Methods This section could use better sub-headings. For example, the first heading, “A mass media campaign in Shandong and Anhui provinces,” could be broken up by “Campaign development” and “Campaign dissemination.” Along these lines, the description of the campaign development and dissemination could be better described. Namely, it was difficult to distinguish whether the authors meant that the campaign concepts/materials were part of the pilot or the actual campaign itself. Moreover, a lot of key details are missing from the text (e.g., how focus groups carried out, etc.). Additional comments are as follows:  -Page 4, Line 10: What organization did the provincial program team members represent, the CDC or Vital Strategies? -Page 4, Line 13: What to the authors mean by “formative research”? Focus groups? If so, would provide more details as to how the focus groups were carried out (e.g., how participants recruited, number of focus groups and members, and when the focus groups were carried out, etc.). - Page 4, Lines 14-15: Why were the age cut-offs for women and men selected? - Page 4, Lines 16-17: Do the authors suspect regional differences exist in the two selected provinces? Why were these two provinces selected? - Page 4, Lines 19-25: Are the communication concepts those that were just piloted during the focus groups? Also, how were these communication concepts presented to formative research/focus group participants? What is via a video, poster, etc? Why were these communication concepts selected to be tested? - Page 4, Lines 25-30: Would clarify the type of data collected, how was it collected, how it was analyzed. - Page 4, Lines 33-35: Would more it more explicit that the results of the formative research informed the development of the health marketing campaign disseminated at the population-level. - Page 4, Line 38: If the campaign was disseminated nationally, why were just two provinces examined? - Page 4, Line 41: Is there a link to the campaign that the authors can include for the readers to view the campaign video? - Page 4, Line 41: What “research demonstrated that an important feature”? - Page 4, Line 45: What do the authors mean by “final script”? Thought it was mentioned earlier that this campaign had been disseminated? - Page 4, Line 53: This should be another sub-section (campaign dissemination or similar). Also, which PSA aired? The “Los with Less Salt” campaign? In the earlier paragraph, it was difficult to understand if the campaigns described were piloted or the actual campaign disseminated itself. -Page 5, Line 8-13: More details are necessary in this section. For example, when did the evaluation take place? Who carried out the evaluation? When authors mention “face-to-face,” do they mean that the survey was administrated by staff? Was the survey piloted prior to use? What language was it available in? How long did it take for respondents to complete? -Page 5, lines 16-35: The authors may consider including a table comparing the sociodemographic breakdown of the sample versus the actual population. Also, -Page 5, line 19: It would help to specify what authors mean by
--	---

“ensure representation across different tiers of society”

- Page 5, line 26: Why was the 25 to 65 year age cut-point chosen?
- Page 5, lines 33-35: This text might be better moved up earlier under the “Evaluation study design” sub-heading
- Page 5, lines 38-53: More details are needed in this section. For example, were questions internally developed or adapted from existing questionnaires? For the campaign awareness measure, it would be ideal to clarify what image shown and for what PSA. Was it the whole video shown, or just screenshots of the video? What question were respondents subsequently asked? Similarly, for “reactions to the campaign,” the authors should not just reference Table 2, but indicate how many agree/disagree statements respondents were asked and how the variable was analyzed.
- Page 6, Lines 3-21: More details on how the variables were measured and operationalized are also needed in this section (e.g., ‘considerations and intentions’ section vague). It may also help to include sub-sub heading for the variables (i.e., dependent variables, independent variables, covariates).
- Page 6, Lines 24-31: Sparse details are included in this section. For example, is it unclear what proportions were compared using chi-squared tests. Was it by province or some other sociodemographic characteristic?
- Page 6, line 28-29: There is a disconnect between what was reported in the measures section versus the variables listed on these lines. For example, why were the ‘consideration and intention’ section not included? In the measures section (above), the authors should also justify why watching television and frequency of internet use were included as covariates.

Results

This section could also use more specificity. For example, on line 51-52, the authors mention “there were significant differences in pre- and post-campaign responses across several measures of demographic characteristics.” What demographic characteristics exactly? In the descriptive results section, would also report percentages.

- Page 68, line 8: Confusing that the authors are reporting intention, behaviors, and barriers. Wasn’t the main analysis carried out to examine the impact of the campaign on knowledge, attitudes, and behaviors?

Discussion

Think that the main issue with the manuscript, as currently written, is that the Introduction does not adequately frame the study purpose. Thus, the rest of the section (including the Discussion) are not entirely clear and align. If the authors can clearly delineate the study aims upfront, then it would help to organize the Discussion (and results) based on each aim.

- Limitations section: The authors should also consider noting campaign recall bias as a limitation. Also, did the authors measure how many times a respondent was exposed to the campaign? It is possible that higher exposure could have led to a higher impact on measures of interest.

Table 1

The table should stand on its own (e.g., add more descriptive title mentioning the campaign, where it was disseminated, etc.). Mention of location (and tiers) comes out of left field as this was never discussed in the text.

	Table 2 Not sure if this table is necessary to include, at least not how the purpose of the study is currently described. This seems to be a sub-analysis table? Also, what do the first 13 self-reported reactions pertain to? What do authors mean by intended interpersonal communication? Table 3 This table is confusing. It might make more sense to just run a pre-post campaign (total) models, and include campaign exposure as a primary regressor. That is, eliminate the unaware and aware model. Also, all of the covariates in the table should be clearly described in the text. Moreover, why are the n's different for "considered following in last 3 months"? Table 4 -Similar to Table 3, this table is also confusing. Unclear why the n's are different for the various variables. Also, how does this fit with the key study aims?
--	--

VERSION 1 – AUTHOR RESPONSE

Comments from Reviewer: 1

Dr. Wei LI, Tianjin Centers for Disease Control and Prevention

The work presented for review raises a very important topic-reducing salt intake. It is important in the world, especially in China, which was the highest salt intake country in the world. High salt intake can lead to high morbidity and mortality of NCD, such as hypertension, kidney diseases, ischemic heart disease and etc. Media investments should be one of the essential way in reducing salt intake Campaign. This manuscript examined the impact of a salt reduction campaign on knowledge, attitudes and behaviours in relation to salt consumption in two provinces of China. It provides guidance for the full implementation of such media campaigns. It is a very meaningful research. I enjoyed reading your study I appreciate the importance of this study. I wrote my concerns to the editor and I am sure s/he will communicate with you. Minor comments:

Comment#1

The comparisons between proportions which were conducted using chi-square tests, whether chi square values can be listed in the table.

Author's response:

We note that the chi-square test indicates whether there is a statistical difference between values of two categorical variables, so the chi-square tests were run to identify significant differences for comparisons of pre-campaign/post-campaign data and comparisons of aware/unaware data for the variables. To report the level of statistical significance of these differences, sp-values are included in the table. This is in line with the practice observed in other papers in BMJ open or other journals. It was also considered that adding chi-square values will clutter the table so these have not been added.

Comment#2

In the "Data analysis" section, why the variate of Location was not be included in the covariates?

Author's response:

The two locations where the survey was conducted were purposively selected since the campaign was only conducted in these two locations. Any possible variations in these two locations have been adequately covered by the socio-demographic variables that have been used as covariates in the model. Hence, using location as a covariate added no additional value after being tested in the model. Additionally, the selection of current covariates was undertaken on the basis of a review of previous literature.

Comment#3

Can the frequency and the ways (TV, billboards, posters and community education activities.....) to get the Campaign information be evaluated in the study?

Author's response:

The data was not captured in a way to perform this analysis.

Comment#4

Page 9 Line 23, "using salt alternatives is a safe way to reduce sodium intake" is a little different with the description in the Table 3. There are some similar expressions. Please try to keep them consistent for the convenience of readers.

Author's response:

We have made the language consistent with the Table to address this suggestion (Page 8, Lines 9-10)

Comments from Reviewer: 2

Dr. Brenda Robles, Division of Chronic Disease and Injury Prevention

The present study sought to examine the impact of a salt reduction sodium campaign of the knowledge, attitudes, and behaviors of residents of two provinces in China. This study is novel in that few manuscripts have focused on sodium reduction, especially on the impact of health marketing campaigns focused on sodium reduction. However, the manuscript can be significantly improved as follows:

Comment#5

Abstract, Objective- is the campaign about salt or sodium reduction? Throughout the main text, "salt" and "sodium" were used interchangeably.

Author's response:

We have added the line "salt is the main source of sodium in our diet" to connect why we are discussing salt intake as being important in reducing sodium consumption (page 3, line 13). We have been careful to not use salt and sodium interchangeably, and have changed several instances where we used "sodium" to instead use "salt."

Comment#6

Abstract, Objective- Also, is this study truly just seeking to examine the impact of the campaign when the majority of the sample was not exposed to it? Would reconsider the study aims to more accurately capture what is presented in the tables.

Author's response:

Yes, the aim of the study is to measure the impact of the campaign. Responses from those who recalled the campaign can usefully be compared with those from respondents who did not recall the campaign to deduce direct campaign effects. In addition, pre- and post-campaign comparisons have been made to assess changes for the total sample, also considering the indirect influence of the campaign through agenda-setting and other channels of message dissemination, including interpersonal channels.

Comment#7

Abstract, Results- do the authors meeting the purchase of "foods" and not "goods"? It would be better to report Cis instead of p-value.

Author's response:

- "Goods" has now been revised to "foods" in the manuscript.
- We note that the chi-square test indicates whether there is a statistical difference between values of two categorical variables, so the chi-square tests were run to identify significant differences for comparisons of pre-campaign/post-campaign data and comparisons of aware/unaware data for variables. To report the level of statistical significance of these differences p-values are reported. This is in line with the practice observed in other papers in BMJ open or other journals. Also, it was considered that adding chi-square values will clutter the table so these have not been added.

Comment#8

Abstract, conclusion- What do the authors mean by encouraging "positive change"? This is a bit vague, would consider eliminating or using more specific language.

Author's response:

We have revised the sentence to be more specific: “to support efforts to reduce salt consumption in China” (Abstract: Page 2, Line 18).

Comment#9, Article Summary

The first bullet is confusing, not sure what the authors mean

Author’s response:

We edited the first bullet to read: “For this study a repeated ‘cross-sectional’ survey design was used rather than a ‘cohort design’ which risks sensitization bias for a mass media campaign evaluation” to make it clearer (Page 2, Lines 23-24).

Comment#10, Introduction

- Found that the Introduction, as currently framed, was a bit disjointed from the rest of the text (i.e., Methods, Results, Discussion). It does not fully set the stage for the study purpose (which is currently unclear). For example, in the Methods section, the authors appear to mention that the campaign was tailored to women because they are the nutritional gatekeepers of the household. If this is the case, the authors should mention this early on in the Introduction and cite pertinent literature to support that women in China are nutritional gatekeepers of the household.
- Also, there some be some mention as to why the two provinces were assessed; is it because the authors suspect there may be regional differences in sodium knowledge, attitudes, and behaviors?
- The Introduction could also be improved some in terms of grammar and syntax.
- Moreover, sodium and salt are used interchangeably (and throughout the text). It might help to provide a definition in the Introduction distinguishing between the two terms.

Author’s Response:

- We added several sentences to set up the background as to why we focused our campaign in the provinces of Anhui and Shandong and aimed to reach women as primary food preparers and men as secondary influencers.
 - o Page 3, Lines 45-46 and Page 4, Lines 1-2: We added “with technical assistance from Vital Strategies, a global public health organization, and the Resolve to Save Lives initiative, which supports governments in both provinces in sodium reduction efforts.”
 - o Page 3, Lines 18-21: We added “Women are the primary preparers of homecooked meals in China and are often the household’s nutritional gatekeepers, while men also exert an influence over food purchase and preparation...”
- We carefully reviewed and edited the introduction for grammar and syntax where it was needed.
- We have added the line “salt is the main source of sodium in our diet” to connect why we are discussing salt intake as being important in reducing sodium consumption (Page 3, Line 13). We have been careful to not use salt and sodium interchangeably, and have changed several instances where we used “sodium” to instead use “salt.”

Comment#11

Line 23: What do authors mean by “physical strength”?

Author’s response:

This mention of “physical strength” refers to research findings indicating the belief that salt contributes to a person being physically strong.

Comment#12

Line 25: The authors should consider providing some examples of “traditional cuisine” dietary preferences in China. Are these “traditional cuisines” high in sodium/salt? If so, would make that more explicit.

Author’s response: We have edited this sentence to discuss preferences for regional traditional cuisine, which in Shandong in particular, tends to be very high in salt (Page 3, Lines 26-29).

Comment#13

Lines 26-27: Would split up this sentence. Something along the lines of, “There is evidence that mass media campaigns can be an effective public health tool for changing social norms and promoting behavior change. For example, studies focused on tobacco cessation have found [XYZ]”

Author’s response: This sentence has been revised as suggested on the above lines (Page 3, Lines 31-34).

Comment#14

Line 49: Would clarify what type of organization is “Vital Strategies” Is it a health marketing firm?

Similarly, what kind of initiative is "Resolve to Save Lives"?

Author's response:

We have added a line indicating that Vital Strategies is a global public health organization and that Resolve to Save Lives, an initiative of Vital Strategies, supports the governments of Shandong and Anhui in sodium reduction efforts. We hope this will also help provide the necessary background on why the mass media campaign was focused on these provinces (Page 3, Lines 45-46 and Page 4, Lines 1-2).

Comment#15

Lines 47-56: This last paragraph can be significantly improved for clarity. While the authors describe the purpose of the campaign, the study purpose/main research question (and corresponding study aims) are unclear. The authors should also briefly describe the research methodology (e.g., cross-sectional, pre-post study carried out in two regions of China?)

Author's response:

The paragraph has been revised to make the study aims and methods clearer (Page 4, Lines 5-7). The last sentence of the last paragraph now reads "The objective of this study is to determine whether the campaign led to changes in knowledge, attitudes, intentions and behaviors around salt use by conducting pre- and post-campaign cross-sectional surveys."

Comment#16, Methods

This section could use better sub-headings. For example, the first heading, "A mass media campaign in Shandong and Anhui provinces," could be broken up by "Campaign development" and "Campaign dissemination." Along these lines, the description of the campaign development and dissemination could be better described. Namely, it was difficult to distinguish whether the authors meant that the campaign concepts/materials were part of the pilot or the actual campaign itself. Moreover, a lot of key details are missing from the text (e.g., how focus groups carried out, etc.). Additional comments are as follows:

Author's response:

This section has been revised to include the "campaign development" and "campaign dissemination" sub-headings. There are now more details provided under each sub-heading, including how focus groups were carried out under the "campaign development" sub-header.

Comment#17

Page 4, Line 10: What organization did the provincial program team members represent, the CDC or Vital Strategies?

Author's response:

The provincial CDC.

Comment#18

Page 4, Line 13: What do the authors mean by "formative research"? Focus groups? If so, would provide more details as to how the focus groups were carried out (e.g., how participants recruited, number of focus groups and members, and when the focus groups were carried out, etc.).

Author's response:

This section has been revised to provide details of the focus group, recruitment, methodology and timing (Page 4, Lines 15-23).

Comment#19

Page 4, Lines 14-15: Why were the age cut-offs for women and men selected?

Author's response:

These gender-specific age cut-offs for the respondents (25- 55 years) were in line with those for the target audience defined for the campaign. An additional group of 56-65 year old women were added to assess any different views from this older age group of mothers/mothers-in-law who may also be influencing cooking practices in the household.

Comment#20

Page 4, Lines 16-17: Do the authors suspect regional differences exist in the two selected provinces? Why were these two provinces selected?

Author's response:

We have now clarified in the introduction that the reason we selected the provinces of Shandong and Anhui are because these are the provinces Vital Strategies' Resolve to Save Lives works in on

sodium reduction efforts (due to high rates of salt consumption and hypertension). That is why these provinces were selected. Analysis of regional differences between the two provinces is beyond the scope of the paper.

Comment#21

Page 4, Lines 19-25: Are the communication concepts those that were just piloted during the focus groups? Also, how were these communication concepts presented to formative research/focus group participants? What is via a video, poster, etc? Why were these communication concepts selected to be tested?

Author's response:

The section has been revised to provide more detail of the presentation of the concepts, and the text describes the production of the campaign based on the research outcomes.

Comment#22

Page 4, Lines 25-30: Would clarify the type of data collected, how was it collected, and how it was analysed.

Author's response:

The quantitative rating and focus group discussion methods have been described for the formative research (Page 4, Lines 15-18), but we believe it is beyond the scope of this campaign evaluation paper (and subject of a separate publication) to describe full details of the methodology and analysis of the formative pre-campaign research.

Comment#23

Page 4, Lines 33-35: Would more it more explicit that the results of the formative research informed the development of the health marketing campaign disseminated at the population-level.

Author's response: The text has been revised to make this clearer (Page 4, Lines 33-36).

Comment#24

Page 4, Line 38: If the campaign was disseminated nationally, why were just two provinces examined?

Author's response: The limited use of a national media channel was chosen as a cost-efficient means to complement the much stronger provincial media where the campaign was primarily targeted and supported by provincial CDC activities. The campaign evaluation focused on the two provinces where the main campaign was conducted.

Comment#25

Page 4, Line 41: Is there a link to the campaign that the authors can include for the readers to view the campaign video?

Author's response: A link will be embedded or URL provided in line with journal specifications for readers to access.

Comment#26

Page 4, Line 41: What "research demonstrated that an important feature"?

Author's response:

We have added the word "formative" in the manuscript before "research demonstrated that an important feature" to make this point clearer (Page 4, Line 38).

Comment#27

Page 4, Line 45: What do the authors mean by "final script"? Thought it was mentioned earlier that this campaign had been disseminated?

Author's response:

This refers to the script of the campaign PSA. We have added "of the PSA" to make it clearer (Page 4, Line 41).

Comment#28

Page 4, Line 53: This should be another sub-section (campaign dissemination or similar). Also, which PSA aired? The "Love with Less Salt" campaign? In the earlier paragraph, it was difficult to understand if the campaigns described were piloted or the actual campaign disseminated itself.

Author's response:

We have created a new sub-section titled "campaign dissemination" and moved any related

information under that header. We rewrote and relabelled the “campaign development” section to make it clearer that one section is focusing on the research behind the development of the campaign, and the other on the final product and how it was disseminated.

Comment#29

Page 5, Line 8-13: More details are necessary in this section. For example, when did the evaluation take place? Who carried out the evaluation? When authors mention “face-to-face,” do they mean that the survey was administrated by staff? Was the survey piloted prior to use? What language was it available in? How long did it take for respondents to complete?

Author’s response:

- The period of evaluation has been mentioned under the sub-section titled “Sampling method” but based your comments we have now shifted it under the sub-section “Evaluation study design” (Page 5, Lines 11-13).
- The evaluation was carried out by an independent research agency hired by Vital Strategies, we have specified that now (Page 5, Line 10). By “face-to-face” we mean in-person surveys, these were done by trained data collectors at the hired independent research agency. We have replaced the term “face-to-face” with the y more common term “in-person” to avoid any confusion (Page 5, Line 10). Yes it was pilot tested and we have added that information (Page 5, Line 31). The questionnaire was in Chinese; this is mentioned in sub-section “Survey questionnaire and measures” (Page 5, Line 31).
- The length of the interviews was approximately 30 minutes, which has been added into the text (Page 5, Lines 33-34).

Comment#30

Page 5, lines 16-35: The authors may consider including a table comparing the sociodemographic breakdown of the sample versus the actual population.

Author’s response:

This has been considered but for certain sociodemographic variables like age and gender, quotas were fixed in the sample to match the target audience of the campaign, reflecting population parameters. We deemed it more critical to present characteristics of the samples from each survey to show the level of consistency between the samples.

Comment#31

Page 5, line 19: It would help to specify what authors mean by “ensure representation across different tiers of society”

Author’s response:

We have edited the sentence to make it clear that we are referencing different tiers of cities: “In each province, cities were selected to ensure representation across different city tiers” (Page 5, Line 18).

Comment#32

Page 5, line 26: Why was the 25 to 65 year age cut-point chosen?

Author’s response:

This age range was selected for the survey sample to align with the age group of the target audience for the campaign. This is reflected in the manuscript (Page 5, Lines 22-23).

Comment#33

Page 5, lines 33-35: This text might be better moved up earlier under the “Evaluation study design” sub-heading

Author’s response: We have moved this text to under the “Evaluation study design” sub-heading.

Comment#34

Page 5, lines 38-53: More details are needed in this section. For example, were questions internally developed or adapted from existing questionnaires? For the campaign awareness measure, it would be ideal to clarify what image shown and for what PSA. Was it the whole video shown, or just screenshots of the video? What question were respondents subsequently asked? Similarly, for “reactions to the campaign,” the authors should not just reference Table 2, but indicate how many agree/disagree statements respondents were asked and how the variable was analysed.

Author’s response:

- For the campaign awareness measure, still images of key scenes from the campaign PSA video were shown to respondents and they were asked whether they had seen the PSA before. This has been made clearer in the manuscript (Page 5, Lines 38-41).

- A battery of 16 statements was used to assess “reactions to the campaign, . This has been added in the manuscript. Analysis of level of agreement was undertaken using a 5-point Likert scale. We have made this clearer in the manuscript (Page 5, Lines 42-45).
- The proportion of respondents indicating “agree” and “strongly agree” on the 5-point Likert scale were aggregated and included in the analysis and reporting. This has been added under sub-section “Data analysis” (Page 6, Lines 17-18).

Comment#35

Page 6, Lines 3-21: More details on how the variables were measured and operationalized are also needed in this section (e.g., ‘considerations and intentions’ section vague). It may also help to include sub-sub heading for the variables (i.e., dependent variables, independent variables, covariates).
Author’s response: This section has been revised in the manuscript to make these methods clearer.

Comment#36

Page 6, Lines 24-31: Sparse details are included in this section. For example, is it unclear what proportions were compared using chi-squared tests. Was it by province or some other sociodemographic characteristic?

Author’s response: Comparisons were for respondents in the pre- and post-campaign surveys and between aware and unaware respondents. This has been added to the manuscript (Page 6, Lines 19-20).

Comment#37

Page 6, line 28-29: There is a disconnect between what was reported in the measures section versus the variables listed on these lines. For example, why were the ‘consideration and intention’ section not included? In the measures section (above), the authors should also justify why watching television and frequency of internet use were included as covariates.

Author’s response: We have updated the text to make what was reported in the “Measures” section and the variables listed in the “Data Analysis” section consistent. Media consumption habits (frequency of watching television and frequency of internet use), as other covariates, is expected to skew results hence were controlled.

Comment#38, Results

This section could also use more specificity. For example, on line 51-52, the authors mention “there were significant differences in pre- and post-campaign responses across several measures of demographic characteristics.” What demographic characteristics exactly? In the descriptive results section, would also report percentages.

Author’s response: In the paragraph following the sentence “there were significant differences in pre- and post-campaign responses across several measures of demographic characteristics,” we describe the differences. We have added percentages to the manuscript to make the differences between the pre- and post-campaign samples clearer (Page 6, Lines 41-45).

Comment#39, Page 68, line 8: Confusing that the authors are reporting intention, behaviors, and barriers. Wasn’t the main analysis carried out to examine the impact of the campaign on knowledge, attitudes, and behaviors?

Author’s response: The evaluation also sought to assess any impact on behavioral intention and perceived barriers to behavior change. This has been made clearer in the manuscript.

Comment#40, Discussion

Think that the main issue with the manuscript, as currently written, is that the Introduction does not adequately frame the study purpose. Thus, the rest of the section (including the Discussion) are not entirely clear and align. If the authors can clearly delineate the study aims upfront, then it would help to organize the Discussion (and results) based on each aim.

Author’s response: The introduction has been revised to more clearly reflect this internal consistency. We think that with those revisions, the introduction and discussion are now well-aligned.

Comment#41, Limitations section: The authors should also consider noting campaign recall bias as a limitation. Also, did the authors measure how many times a respondent was exposed to the campaign? It is possible that higher exposure could have led to a higher impact on measures of interest.

Author’s response: Reference to recall bias has been included in the limitation section (Page 10, Lines 37-38). The study did not measure frequency of exposure to the campaign and this is stated in

the manuscript (Page 10, Line 34).

Comment#42, Table 1

The table should stand on its own (e.g., add more descriptive title mentioning the campaign, where it was disseminated, etc.). Mention of location (and tiers) comes out of left field as this was never discussed in the text.

Author's Response: The title for Table 1 has been revised. City tiers have now been explained in the manuscript text in relation to sampling (Page 5, Line 18). The location has been added to sociodemographic information in the manuscript.

Comment#43, Table 2

Not sure if this table is necessary to include, at least not how the purpose of the study is currently described. This seems to be a sub-analysis table? Also, what do the first 13 self-reported reactions pertain to? What do authors mean by intended interpersonal communication?

Author's Response: Findings from Table 2 are described in the results section under the header "Reactions to the campaign." It captures how those who were aware of the campaign in the post-campaign survey reacted to it. We think it is important to include because it reports on how the campaign PSA was perceived by those who saw it, and is discussed in detail in the text. We have added a line to explain the findings on interpersonal communication under the "Reactions to the campaign" heading in the results section (Page 7, Lines 20-21). It is important to assess this ability for a campaign PSA to generate further message dissemination through interpersonal communication amongst those who saw it. This has been clarified in the manuscript.

Comment#44, Table 3

This table is confusing. It might make more sense to just run a pre-post campaign (total) models, and include campaign exposure as a primary regressor. That is, eliminate the unaware and aware model. Also, all of the covariates in the table should be clearly described in the text. Moreover, why are the n's different for "considered following in last 3 months"?

Author's Response: To answer the objective of this paper we defined the analysis plan to compare aware versus unaware groups and have run the models accordingly. This is based on the literature and is what we have done when reporting our campaign results in previous papers. Some questions were asked only to those who prepare food in the household. Hence, the "n" for such questions is different; this question is one of them.

Comment#45, Table 4

Similar to Table 3, this table is also confusing. Unclear why the n's are different for the various variables. Also, how does this fit with the key study aims?

Author's Response: Table 4 contains the key findings of comparisons for pre-and post-campaign measures and aware/unaware comparisons on intentions, behaviours and barriers to the changes, which address the study' aims of assessing these changes following the campaign. Some questions were asked only to those who prepare food in the household. Hence, the "n" for such questions is different; this question is one of them.